# Preparation and Characterization of Graphene Oxide/Polyaniline/Polydopamine Nanocomposites towards Long-Term Anticorrosive Performance of Epoxy Coatings

**DOI:** 10.3390/polym14163355

**Published:** 2022-08-17

**Authors:** Yunyun Huang, Borui Zhang, Jian Wu, Ruoyu Hong, Jinjia Xu

**Affiliations:** 1College of Chemical Engineering, Fuzhou University, Quanzhou 362000, China; 2Department of Chemistry and Biochemistry, University of Missouri—St. Louis, One University Boulevard, St. Louis, MO 63121, USA

**Keywords:** polydopamine, polyaniline, graphene oxide, anti-corrosion, polymer coatings

## Abstract

To address the challenging issues of metal materials corrosion in industries, which has caused huge economic losses and security threats to many facilities in marine environments, functional polymer coatings have been widely used and regarded as one of the simplest and most effective methods to prevent such an undesirable event. In this study, a new type of coating filler consisting of graphene oxide/polyaniline/polydopamine (GO-PANI-PDA) nanocomposites has been successfully synthesized. The morphology, structure, composition, and corrosion resistance performance of the GO-PANI-PDA (GPP) nanocomposites were investigated via a series of characterization methods. The results from our electrochemical impedance spectroscopy, potentiodynamic polarization curve and salt spray experiment showed that the best corrosion resistance performance of the coating is from GPP 21 with the epoxy/GO-PANI:PDA ratio of 2:1, which exhibited a positive corrosion potential (−0.51 V) shift from epoxy/GO-PANI coating (−0.64 V). The corrosion current density (3.83 × 10^−8^ A/cm^2^) of GPP 21 is nearly an order of magnitude lower than that of epoxy/GO-PANI (7.05 × 10^−7^ A/cm^2^). The good anti-corrosion performance was fascinatingly observed in salt spray tests even without obvious corrosion phenomenon after 30 days of testing. Due to these remarkable properties, GPP nanocomposites can be an outstanding candidate for the rapid development of broadband shielding and anticorrosive materials.

## 1. Introduction

Over past few decades, many effective strategies such as adding corrosion inhibitors, cathodic protection, surface treatment and coating technology have been widely developed to prevent corrosion in the marine industry and ensure the structural integrity of the alloy [1,2,3]. Among them, coating technology is the most widely used anti-corrosive approach, not only because it has a simple process, but also the low porosity generated from the coating can effectively retard corrosion [4,5].

Epoxy resin (EP) has good mechanical properties, considerable adhesion and stable chemical properties. However, corrosive media such as H_2_O, O_2_ and Cl^−^ will inevitably penetrate into its coating layers and reach the substrate surface through holes and defects formed by solvent volatilization during long-term immersion [6]. One of the most effective ways to solve this problem is to dope nanomaterials to epoxy resins, where nanomaterials can effectively prevent those corrosive medium from entering the coating layers through the filling of micropores. Many efforts have been denoted to introduce nanomaterials such as clay, carbon nanotubes, TiO_2_, SiO_2_ and graphene into epoxy coatings to improve their anti-corrosion properties [7,8,9,10,11,12]. Among these materials, graphene oxide (GO) has a the characteristics of high specific surface area, good mechanical strength and physical barrier effect, which can significantly prolong the diffusion path of corrosive medium [13,14,15,16,17,18]. Polyaniline (PANI) is a well-known conductive polymer, which has light-weight, non-toxic, low-cost, good electrochemical and reversible redox characteristics [19,20,21]. Many reports have confirmed that PANI coating with a certain thickness has the effect on physical shielding and the special redox performance, which can promote the formation of a protective oxide film on the surface of metal and passivates the metal matrix [22,23]. DeBerry et al. [24] found that PANI as an additive can alleviate the corrosion of metals. Schauer [25] and Zhang [26] added PANI to epoxy resin to form a good nanocomposite, where PANI can generate electrons to form a passivation film and form a chemical shielding layer [27]. Although GO-PANI nanocomposite can improve its barrier properties for oxygen and water with the covalent grafting of PANI and GO, the aggregation of nanocomposites will produce undesired pores and defects, which will in turn accelerate the penetration rate of corrosive medium and eventually lead to serious corrosion.

The amino and hydroxyl functional groups of dopamine (DA) make it easy to be functionalized and cross-linked with other materials, resulting in the improvement of the dispersion ability and the compatibility between fillers and resins [28,29,30,31,32,33]. Benefitting from good compactness, anchorage and uniformity, the polydopamine (PDA) coating is a promising candidate filler for an anti-corrosion coating [34,35]. Zhu et al. [36] found that DA can enhance the compatibility and adhesion of graphene with epoxy resins and avoided galvanic corrosion. Haruna et al. [37] found that dopamine-functionalized graphene oxide (DA-GO) formed a protective layer to prevent carbon steel from corrosion. Due to the hydrogen bond between GO and PDA as well as the electrostatic interaction between -COO^−^ in GO and -NH^3+^ in PDA, we propose that the grafting of dopamine on GO will help promote the compatibility of GO-PANI nanosheet in epoxy resin, thereby improving the corrosion inhibition efficiency of the coating. Nastaran et al. [38] prepared a novel nanocomposite Ag-Pd-PDA/RGO to reduce the consumption of palladium, improve the catalytic activity of Pd, and have a synergistic effect. Bahram et al. [39] confirmed that both barrier and inhibitive action of epoxy coating were significantly enhanced in the presence of GOQD-PANI. The fine GOQD-PANI well dispersed in the epoxy matrix, filled the pores and defects and blocked the diffusion pathways.

In this paper, GO-PANI nanosheets were functionalized with PDA to fabricate GO-PANI-PDA(GPP) nanocomposites by in situ chemical polymerization of dopamine (DA) in alkaline environment. The structure and morphology of GPP nanocomposites were characterized by scanning electron microscope (SEM), Fourier-transform infrared spectroscopy (FT-IR), and X-ray diffraction (XRD) measurements. The anti-corrosion performance of EP/GPP coatings were tested by electrochemical methods and salt spray experiments. The mechanical properties, hydrophilic and hydrophobic properties, and adhesion properties of the coatings were tested by impact resistance, contact angle and adhesion tests. The GO-PANI-PDA (GPP) coating we prepared here will not only play the sheet barrier role of GO prolongs the diffusion path of corrosive substances, but also avoid forming galvanic corrosion.

## 2. Experimental Section

### 2.1. Materials

GO was purchased from Shenzhen Tulingjinhua Tech. Co., Ltd. (Shenzhen, China). Dopamine was procured from Bomei Biotechnology Co., Ltd. (Hefei, China). Xylene was obtained from Yongbang Chemical Technology Co., Ltd. (Jinan, China). Acetone was supplied from Jiujia Chemical Co., Ltd. (Suzhou, China). Concentrated hydrochloric acid (HCl) was obtained from Shenzhen Qihongyuan Technology Co., Ltd. (Shenzhen, China). Sodium chloride was procured from Yatai United Chemical Co., Ltd. (Wuxi, China), Ethanol (C_2_H_5_OH, 99.5%) was obtained from Linshi Chemical Reagent Co., Ltd. (Guangzhou, China). Aniline (AN) and ammonium persulfate (APS) were purchased from Sinopharm Chemical Reagent Co., Ltd. (Shanghai, China). Epoxy resin (E−44) and curing agent (polyamide 650) were supplied from Liangli Electronic Commerce Co., Ltd. (Shanghai, China). The deionized water was used throughout the experiments.

### 2.2. Preparation of GO-PANI-PDA Nanocomposites

The synthetic route of GO-PANI-PDA(GPP) nanocomposites was shown in Figure 1. An amount of 20 g of GO solution was added to a beaker. Then, 3 g of aniline and 150 mL of dilute hydrochloric acid (1 mol/L) were added into the GO solution. The mixture was stirred for 10 min and ultrasonicated for 20 min. The GO-AN suspension was poured into a three neck flask (Chengyi Nuohua Glass Instrument Co., Ltd., Guangzhou, China) in an ice bath at −5~0 °C and stirred evenly. Subsequently, 50 mL of APS solution (0.1 g/mL) was added dropwise to the mixture and stirred for 24 h. Finally, the product was centrifuged and washed with deionized water and anhydrous ethanol several times, then dried for 24 h, at 60 °C, in a vacuum oven (Hesheng Instrument Technology Co., Ltd., Shanghai, China). The product of GO-PANI nanocomposites was obtained.

Then, 250 mL deionized water was added in a 250 mL beaker, followed by adding 1500 mg of GO-PANI and PDA in solution and stirring for 12 h (pH 8.5 in tris buffer solution). After standing for 24 h, the GO-PANI-PDA (GPP) nanocomposites were obtained by filtering and washing with deionized water and anhydrous ethanol several times, and dried under vacuum, at 60 °C, for 24 h. The products with varying weight ratio of 4:1, 2:1, 1:1, 1:2 and 1:4 between GO-PANI and PDA were prepared and named as GPP 41, GPP 21, GPP 11, GPP 12 and GPP 14, respectively. The control sample PDA without incorporating GO-PANI was also prepared for comparison.

### 2.3. Coating Preparation

The procedure for coating preparation was shown in Figure 2. An amount of 125 mg of GPP 41 was added into 3 mL of xylene and sonicated for 60 min. Then, 3 g of curing agent (polyamide 650) was added to the mixture and stirred thoroughly for 7 min at 1800 rpm. Additionally, then 3 g of epoxy resin was added to the mixture and stirred another 5 min at 1800 rpm. The mixture was centrifugated to remove bubbles [37] and then coated on the surface of Q 215 steel home-made electrode (exposed area of 1 cm^2^) and plate with a wire bar coater (200 μm and 100 μm) and cured for 24 h and then dried, at 60 °C, for 24 h to give corresponding coatings. It should be noted that the working electrode was polished with 500-mesh, 1000-mesh and 2000-mesh sandpapers in sequence and washed with alcohol to obtain clean and smooth surface before coatings. The coatings by GO-PANI, PDA, GPP41, GPP21, GPP11, GPP12 and GPP14 were prepared similarly and noted as EP/GP, EP/PDA, EP/GPP41, EP/GPP21, EP/GPP11, EP/GPP12 and EP/GPP14, respectively.

### 2.4. Characterization

The surface morphology of the prepared materials was characterized by scanning electron microscope (SEM) (Thermo Fisher Scientific, Waltham, MA, USA) and the surface functional groups were analyzed by Fourier-transform infrared (FT-IR) spectroscopy with a scanning range of 4000~500 cm^−1^. The crystal structures of prepared materials were characterized by X-ray diffraction (XRD) (Panalytical, Beijing, China)with a PANalytical X′Pert Pro X-ray diffractometer. The electrochemical impedance spectroscopy (EIS) and potentiodynamic polarization measurements were adopted to detect the corrosion protection performance of each coatings in 3.5% NaCl solution, at room temperature. EIS tests were carried out at an CS165 electrochemical workstation (CorrTest Instruments Corp., Ltd., Wuhan, China) with a three-electrode system including the working electrode (steel electrode), the counter electrode (platinum electrode) and reference electrode (Ag/AgCl Electrode). The frequency was in the range of 10^5^–10^−2^ Hz with an amplitude of ± 5 mV. The potentiodynamic polarization curves were obtained with the sweep rate of 0.5 mV/s. The corrosion resistance test of the coatings was carried out for continuous 30 days with the salt-spray test chamber (FQY-015, Modern Environment Engineering Technique Co., Ltd., Shanghai, China) [35]. The adhesion of the coatings was tested by the cross hatch tester (BGD 502/2, Biuged Laboratory Instruments Co., Ltd., GuangZhou, China). The conductivity was tested with a conductivity tester (ST-2722, Jingge Electronic Co., Ltd., Suzhou, China). The hydrophilicity and hydrophobicity of the coatings were tested by a contact angle measuring instrument (CA-100C, innuo precision instrument Co., Ltd., Shanghai, China).

## 3. Results and Discussion

### 3.1. Morphology Analysis

To analyze the microstructure of the prepared nanocomposites, SEM images were taken. Figure 3a,b show SEM images of pristine GO and PANI, which are sheet-like aggregates and a fine fiber structure, respectively. After in situ polymerization on the surface of graphene, the diameter and the length of the GO-PANI becomes thicker and shorter as shown in Figure 3c, which is consistent with the reported literature [40]. The PDA materials were found to be spherical particles [41] (Figure 3d). Interestingly, the morphological changes were observed significantly as the PDA-to-GO-PANI weight ratio changes. When the ratio is 1:2, button-like PDA aggregate appears on the surface of GO-PANI, while when the ratio increases to 2:1, the GPP21 shows a cross-linked structure composed of more smaller nanoparticles, as displayed in Figure 3e,f. This result is probably due to the deposition of PDA that has strong adhesion on the surface of GO-PANI.

Some pores and agglomeration defects were present in the epoxy coatings filled with GO-PANI (GP) in Figure 4a, which is attributed to the poor compatibility of GO-PANI and epoxy resin. Therefore, GO-PANI nanocomposites cannot perfectly exert to the inherent barrier performance and corrosion resistance. On the contrary, modification of PDA on GO-PANI might help to achieve uniform dispersion of GO-PANI in epoxy resin as it is indicated from the fact that the fracture surface of epoxy coatings filled with GO-PANI-PDA nanocomposites (Figure 4b) is uniform without agglomeration, displaying the good compatibility and laying a solid foundation for corrosion resistance.

### 3.2. XRD Analysis

As presented in Appendix A of the XRD patterns of GO, PANI and GO-PANI, the additional diffraction peaks for GO-PANI was observed, indicating that the successful integration of GO with PANI. The XRD patterns of GO-PANI and GO-PANI-PDA nanocomposites are shown in Figure 5. It can be seen that the GO-PANI have four characteristic peaks at 2*θ* = 15.2°, 20.3°, 25.2°, and 26.5°, corresponding to (011), (020), (200) and (121) planes, respectively. The XRD pattern of GO-PANI-PDA nanocomposites is similar to that of GO-PANI, which proves that the modification of PDA does not change the crystal structure of GO-PANI [42,43,44]. Additionally, the intensity of the diffraction peak of GO-PANI-PDA nanocomposites is drastically reduced, indicating that PDA has been successful integrated on the surface of the GO-PANI and has great interface interaction.

### 3.3. FT-IR Analysis

The FT-IR spectra of PDA, GO, GO-PANI and GO-PANI-PDA are shown in Figure 6 and Appendix A. The bands found at 1505 cm^−1^ and 1121 cm^−1^ are related to the stretching vibration of N-H bond and C-N bond, respectively. The peak at 3411 cm^−1^ is caused by the O-H stretching vibration of GO-PANI-PDA [33]. The C=C stretching vibration in PDA is found at 1615 cm^−1^, which shifts to 1587 cm^−1^ for GO-PANI-PDA nanocomposite. Moreover, the characteristic peaks of PDA observed at 1343 cm^−1^ and 1587 cm^−1^ also appear in the FT-IR spectrum of GO-PANI-PDA. All these results indicates that PDA has been successfully attached to GO-PANI.

### 3.4. Electrochemical Measurements

#### 3.4.1. The Electrochemical Impedance Spectroscopy

The electrochemical impedance spectroscopy (EIS) and potentiodynamic polarization curves were applied to investigate the corrosion resistance of EP/GP, EP/PDA, EP/GPP41, EP/GPP21, EP/GPP11, EP/GPP12 and EP/GPP14 coatings. The reference electrode was an Ag/AgCl electrode, which was calibrated before use to ensure that the same reference potential for each experiment. The working electrode was placed in 3.5 wt% NaCl solution for 3600 s to test the stability of the open circuit potential [45]. When the potential fluctuation did not exceed 0.5 mV in the last 10 min, electrochemical impedance spectroscopy and potentiodynamic polarization curves were measured using an electrochemical workstation [46,47]. The experiment was carried out twice to ensure the reproducibility of different samples, and then ZSimpWin software was used to fit and analyze the EIS data.

Figure 7 shows the EIS results of different coatings prepared by EP, EP/PDA, EP/GP and EP/GPP nanocomposites. In Nyquist plots, the imaginary part of the impedance is related to the real part by semicircles. The height and width of these semicircles are related to the capacitive and resistive behavior of the coating, respectively. The diameter of the semicircle represents the total resistance of the coating, while a resistance value is too small, meaning that severe localized corrosion may have occurred [48,49,50]. Figure 7a shows that the addition of GO-PANI-PDA increased the diameter of the semicircular ring, leading to the high corrosion resistance of the EP/GPP coating. It can be seen that the corrosion resistance of coatings is significantly affected by the weight ratio between GO-PANI and PDA. When the ratio of PDA to GO-PANI increases from 1:4 to 1:2 (GPP41 to GPP21), the anti-corrosion performance of the coating was greatly improved, while it turned to decline with the further increase the ratio of PDA. Interestingly, we found that the EP/GPP 21 coating has the best anti-corrosion performance among all coatings. The impedance of the coatings undergoes a similar trend as shown in Figure 7b, indicating that the corrosion inhibition effect of EP/GPP increased first and then decreased with the increasing weight ratio of PDA to GO-PANI.

Figure 8 displayed the equivalent circuit model of the EP/GPP 21 coatings in 3.5 wt% NaCl solution, where R_e_ is the solution resistance/electrolyte resistance, R_po_ is the coating resistance, R_ct_ is the charge transfer resistance, C_ζ_ is the coating capacitance, C_dl_ is the electric double-layer capacitance/film interface constant phase element [32], msd represents the measured value, and cal represents the fitted value. It is well known that a large value of R_ct_ of coating corresponded to good corrosion resistance. The value of C_ζ_ can reflect the permeability resistance of the coating and the value of C_dl_ corresponds to the defects in coatings, and the wet adhesion between the coating layer and the metal. For the EP/GPP 21 coating, the values of R_ct_ and R_po_ reached 5.279 × 10^16^ Ω·cm^2^ and 3.183 × 10^6^ Ω·cm^2^, respectively, indicating the coating has good corrosion resistance, whereas the values of C_ζ_ and C_dl_ are as small as 7.257 × 10^−10^ Ω·cm^2^ and 6.071 × 10^−9^ Ω·cm^2^, respectively, indicating that a dense layer of PDA pre-formed on the surface of the substrate. It can be reasonably concluded that GPP nanocomposites can fill the defects of epoxy resin in addition to the physical barrier effect of the coating with the adhesion properties of PDA.

#### 3.4.2. Potentiodynamic Polarization Curves

The polarization experiments of EP/GPP nanocomposite coatings in 3.5 wt% NaCl solution were carried out. The corresponding Tafel curves and specific data are shown in Figure 9 and Table 1. Compared with the anodic and cathodic polarization curves of EP/GP coatings, the curves of EP/GPP coatings are shifted positively, which means that EP/GPP coating exhibits high corrosion potential during anodic and cathodic polarization. The cathodic and anodic branches are extrapolated to their intersections to obtain the corrosion current (Icorr), the corrosion potential (Ecorr) and the corrosion rate (CR) [51]. The corrosion potential of EP/GPP21 coating (−0.51 V) is shifted to the positive direction than that of EP/GP coating (−0.64 V). Its corrosion current density (3.83 × 10^−8^ A/cm^2^) is also nearly an order of magnitude lower than that of EP/GP (7.05 × 10^−7^ A/cm^2^), and the coating exhibits the smallest current density during cathodic polarization, corresponding to the corrosion rate of 4.50 × 10^−4^ mm/year is also an order of magnitude lower than that of EP/GP of 8.27 × 10^−3^ mm/year. These results indicated that the passivation zone was formed and EP/GPP21 coating has the best corrosion resistance. The similar results from Tafel analysis also suggested the same conclusion as that obtained by the EIS analysis.

The low values of Icorr and Ecorr exhibited by EP/GPP can be attributed to the surface-adsorbed GO-PANI-PDA molecules, and the addition of uniformly dispersed GO-PANI-PDA nanosheets can compensate for the defects in the epoxy resin matrix and improve the dispersibility and interfacial adhesion of GO-PANI nanoparticles to epoxy resin. It also further prevents the steel from being corroded by corrosive medium through the “labyrinth effect” of corrosive ion penetration, which significantly reduces the CR of GO-PANI-PDA and improves the corrosion resistance of the coating. As a result, the EP/GPP21 has a highest corrosion inhibition efficiency. It is worth noting that the thickness of the electrode coating is very critical when making the electrode coating, and finding a more accurate coating thickness control method will be of great help to the experiment.

#### 3.4.3. Salt Spray Tests

For the salt spray tests, each sample was coated with two thicknesses of 100 μm and 200 μm, and placed in a salt spray test box after edge sealing. Figure 10 showed the optical images of the panels coated with EP/GP, EP/PDA, EP/GPP14, EP/GPP12, EP/GPP11, EP/GPP21 and EP/GPP41 over 7 days after salt spray. We saw the occurrence of corrosion in all samples, especially for EP/GP, EP/PDA, EP/GPP12, EP/GPP14 coatings shown in A, B, F, and G of Figure 10. Their distribution of corrosion products is greater than other coatings having obvious corrosion and rust. However, in the case of EP/GPP41, EP/GPP21, and EP/GPP11coatings shown in C, D, and E in Figure 10, the surface was relatively clean with less corrosion lines and reduced corrosion degree. Among them, the surface of the EP/GPP21 coated steel plate (Figure 10D) exhibited the smoothest surface without corrosion product stripes detected, indicating that EP/GPP21 coating has the best corrosion resistance.

Further salt spray experiments were performed on the EP/GPP21 coatings with different thickness (100 μm and 200 μm). The results over the salt spray test duration are as shown in Figure 11. It should be noted that we did not observe any significant thickness changes in coatings during the salt spray experiments. The earliest corrosion occurred in the left part of the plate in Figure 11 (D7), and the corresponding corrosion time was 35 days with 100 μm thickness of EP/GPP21 coatings. Surprisingly, the surface with 200 μm thickness of EP/GPP21 coatings (Figure 11, D8) did not exhibit such corrosion, indicating that the 200 μm thickness of EP/GPP21 coatings has better anti-corrosive performance under the same salt spray testing conditions. The salt spray experiment was continued until 45 days. As it can be seen in the Figure 11, the corrosion area of the 100 μm plate expanded, and the right side of 200 μm gradually corroded and rusted. This is because the corrosive medium contacts the substrate after the coating is damaged, which further accelerates the corrosion of the metal, but the corrosion part of the two plates is less than 30%. The sample continued to corrode in the salt spray environment for 35 days. We found that EP/GPP21 composite coating has the best anti-corrosion properties, indicating that the well-dispersed GO-PANI-PDA nanosheets are beneficial to improve the anti-corrosion performance of epoxy coatings. This corrosion resistance performance is attributed to the strong adhesion performance of PDA between the coating and the steel, and fully exerts the dense barrier performance and electrochemical redox performance of the EP/GP coating structure.

### 3.5. Adhesion Test

Next, to evaluate the adhesion performance of the coating, we applied sufficient force to completely break the coating and check the scratch on the surface of the substrate to see whether the nearby coating falls off. Figure 12a,b show the optical images of the EP/GPP coating before and after cross-cut stripping experiments. After the EP/GPP is peeled off by the tape, the grid does not appear to fall off or leave residues, indicating that the coating has a good adhesion. According to the ASTMD3359-97 standard, the adhesion level of our coating reaches the level of 5A. This is because the catechol group of the PDA coating has a strong metal coordination ability, which can promote the in situ deposition of metal oxides under the coating on the surface of the material, and the good binding ability improves the interaction between the nano-hybrid material and the metal, giving rise to a reasonable stability.

### 3.6. Conductivity Measurement

It can be seen from Figure 13 that the resistivity of GO-PANI and PDA is 0.227 Ω·cm^−1^ and 803 Ω·cm^−1^, respectively. After polymerization, the resistivity of GO-PAN-PDA is close to that of PDA, which is much larger than that of GO-PANI. This result suggests that the addition of PDA to GO-PANI not only maintains the sheet-like barrier properties of GO-PANI, but also increases the resistivity of the material and reduces the corrosion current, thereby enhancing the anti-corrosion performance of the composite. However, when the amount of doped PDA is small, the resistivity is relatively large. It can be reasonably explained that the excessive addition of PDA in the anti-corrosion performance test decline the anti-corrosion performance. It is worth noting that although the less PDA is added, the higher the resistivity is, a certain amount of PDA is also required to support the compatibility and adhesion between materials.

### 3.7. Contact Angle

The hydrophilicity and hydrophobicity of the coating surface were determined by contact angle in the presence of water. As shown in Figure 14a,b, the contact angles of the EP/PDA coating and the EP/GPP coating were 81.31° and 89.64°, respectively, indicating that the hydrophobicity was greatly improved after the integration of PDA to GO-PANI. With the water-resistant barrier effect enhanced, the increased hydrophobicity of the EP/GPP coating could be beneficial for preventing the penetration of corrosive medium, leading to its good performance in the corrosion resistance test.

## 4. Conclusions

GO-PANI-PDA nanocomposites were successfully synthesized as an ideal filler to improve the anticorrosion properties of epoxy coatings. The GO-PANI-PDA nanocomposites were characterized by FT-IR, SEM and XRD, which proved that the PDA and GO-PANI were completely incorporated without changing the structure of GO-AN nanocomposites. The corrosion resistance behavior and mechanism of the EP coatings filled with GO-PANI-PDA nanocomposites were systematically studied by EIS, Tafel polarization curves and salt spray experiments. The results showed the EP/GPP21 coating had the best corrosion resistance with the positive corrosion potential (−0.51 V) shifted compared with that of the EP/GP coating (−0.64 V). The corrosion current density (3.83 × 10^−8^ A/cm^2^) was also nearly an order of magnitude lower than that of EP/GP (7.05 × 10^−7^ A/cm^2^), which corresponded to the lowest corrosion rate of 4.50 × 10^−4^ mm/year. The salt spray test displayed that EP/GPP 21 coating had over 30-day long-term preservative effect on Q215 steel. Doping GO-PANI in combination with the excellent properties of PDA gives GO-PANI nanosheets better compatibility and dispersion in epoxy resins, strengthening the bonding performance between the coating and the substrate. It can also adjust the electrical conductivity of the material, fill the defects in EP/GP and improve the barrier properties and electrochemical passivation properties of GO-PANI. This work is expected to be widely used in the field of long-term corrosion protection, providing the necessary protection for various marine installations, ships and other metal components.

## Figures and Tables

**Figure 1 polymers-14-03355-f001:**
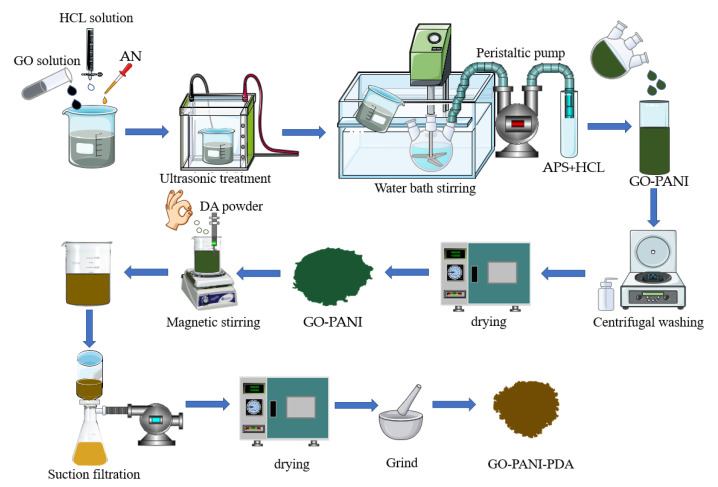
The schematic diagram of GO-PANI-PDA nanocomposites preparation.

**Figure 2 polymers-14-03355-f002:**
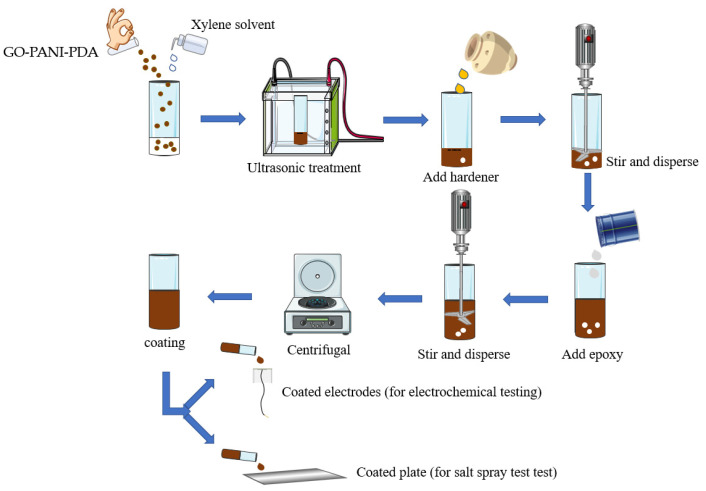
The schematic diagram of the preparation of GO-PANI-PDA nanocomposites coating.

**Figure 3 polymers-14-03355-f003:**
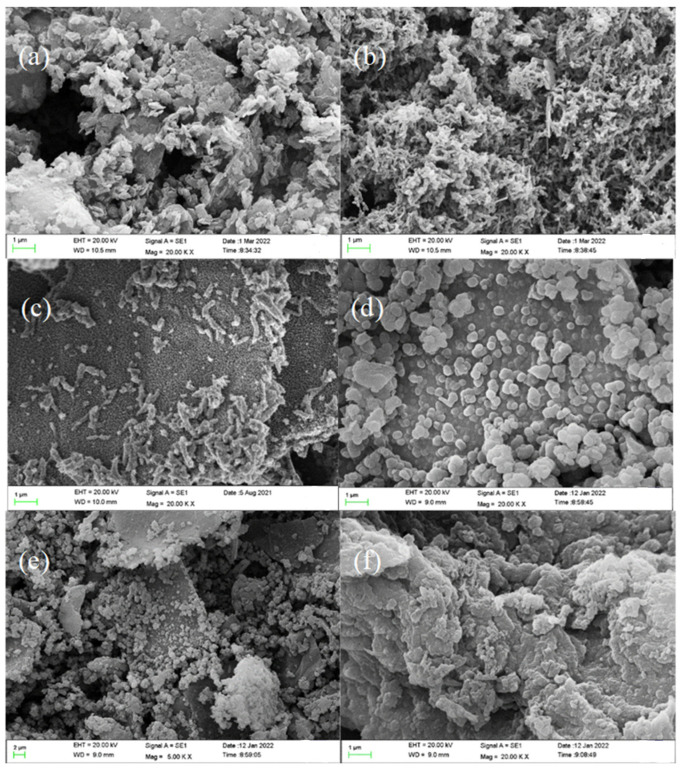
SEM images of (**a**) GO, (**b**) PANI, (**c**) GO-PANI, (**d**) PDA, (**e**) GPP 21, and (**f**) GPP 12.

**Figure 4 polymers-14-03355-f004:**
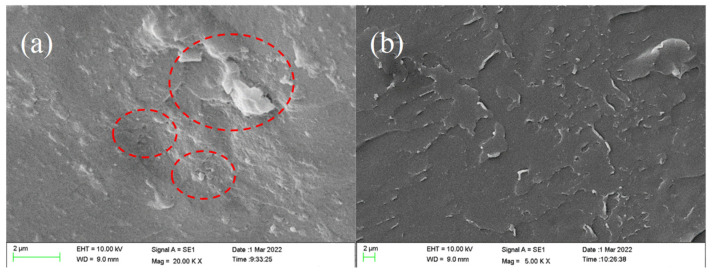
SEM images of (**a**) EP/GP and (**b**) EP/GPP.

**Figure 5 polymers-14-03355-f005:**
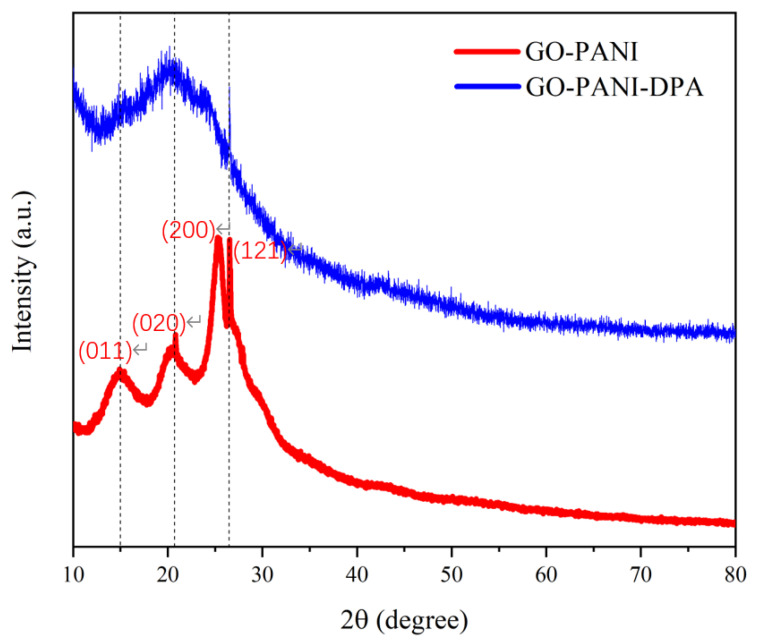
XRD patterns of GO-PANI and GO-PANI-PDA nanocomposites.

**Figure 6 polymers-14-03355-f006:**
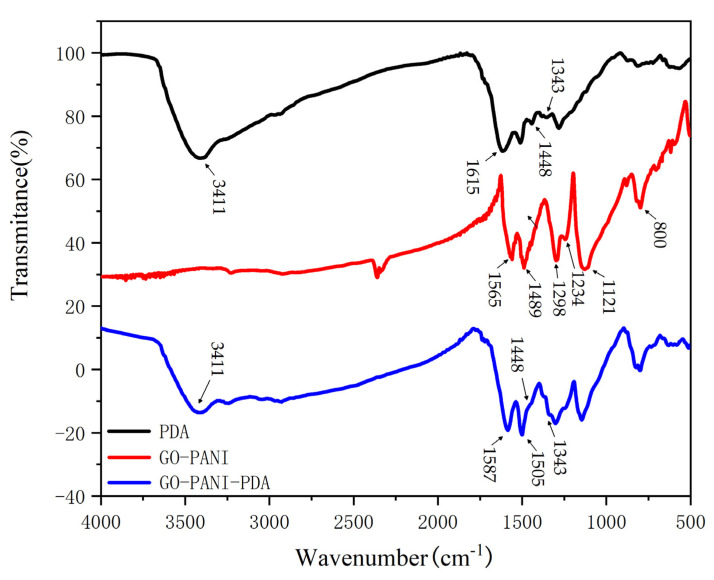
FT-IR spectra of GO-PANI, PDA and GO-PANI-PDA nanocomposites.

**Figure 7 polymers-14-03355-f007:**
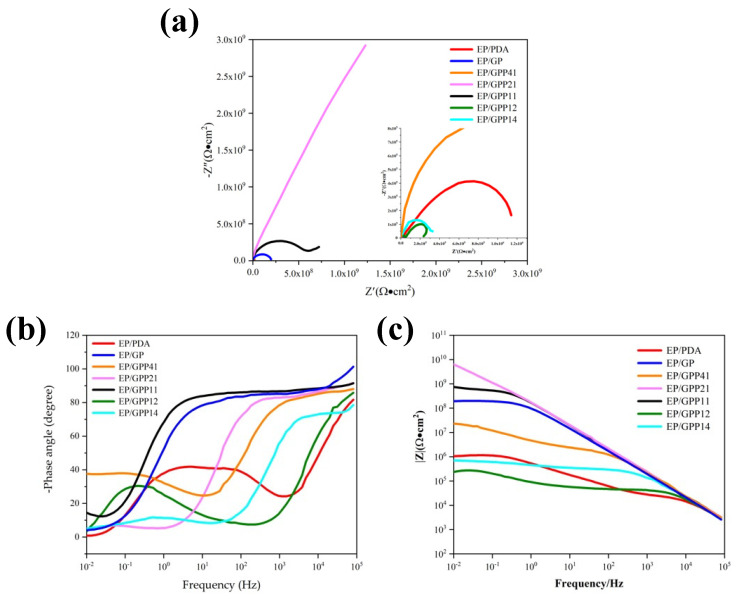
Nyquist and Bode diagrams of the EP, EP/PDA, EP/GP and EP/GPP nanocomposites coatings: (**a**) Nyquist plots, (**b**) Bode plots of |Z|, and (**c**) Bode plots of phase angle.

**Figure 8 polymers-14-03355-f008:**
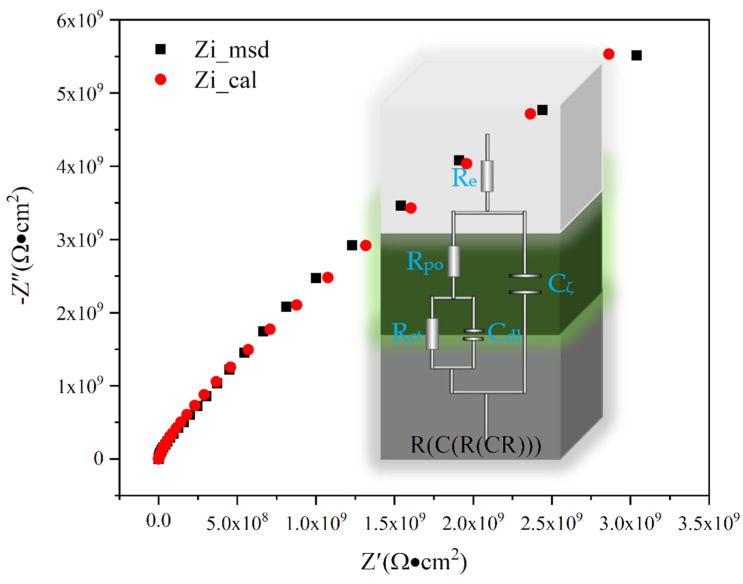
Analysis of the EIS data using an equivalent circuit model.

**Figure 9 polymers-14-03355-f009:**
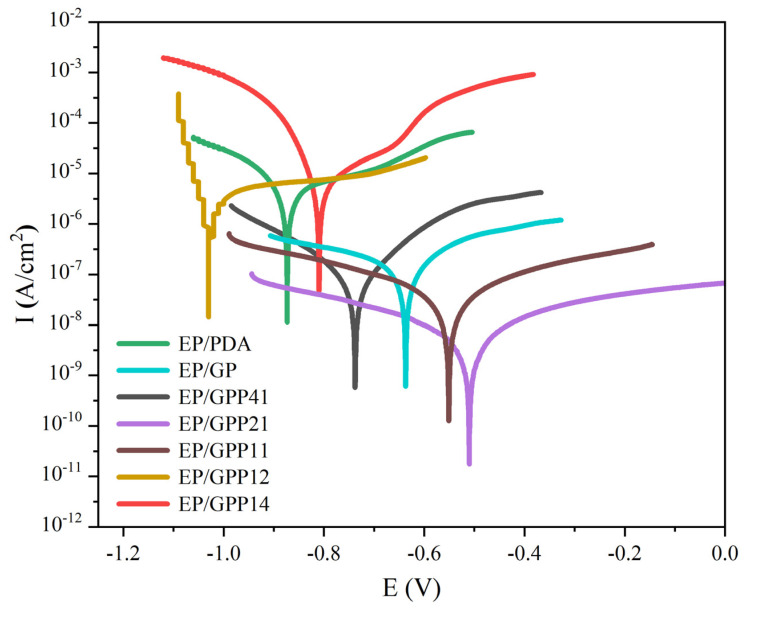
Potentiodynamic polarization curves for the different coatings.

**Figure 10 polymers-14-03355-f010:**
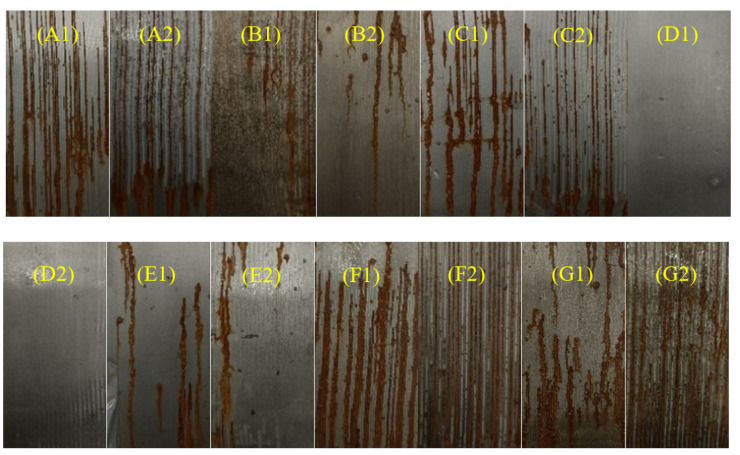
Photographs of the panels coated with (**A**) EP/GP, (**B**) EP/PDA, (**C**) EP/GPP41, (**D**) EP/GPP21, (**E**) EP/GPP11, (**F**) EP/GPP12 and (**G**) EP/GPP14 over 7 days after salt spray.

**Figure 11 polymers-14-03355-f011:**
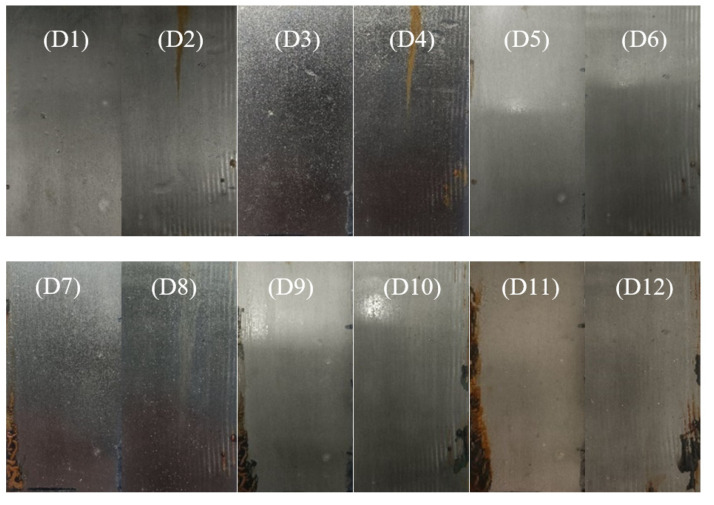
Photographs of EP/GPP21 coating for 20–45 days salt spray test. D1 (20 d, 100 μm), D2 (20 d, 200 μm), D3 (25 d, 100 μm) ), D4 (25 d, 200 μm), D5 (30 d, 100 μm), D6 (30 d, 200 μm), D7 (35 d, 100 μm), D8 (35 d, 200 μm), D9 (40 d, 100 μm), D10 (40 d, 200 μm), D11 (45 d, 100 μm), D12 (45 d, 200 μm).

**Figure 12 polymers-14-03355-f012:**
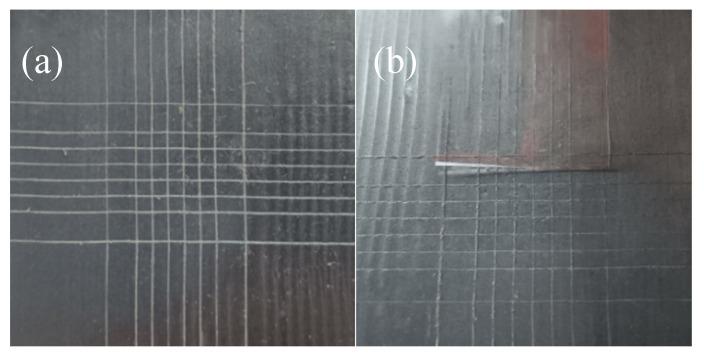
The photograph of the EP/GP and EP/GPP coating as-cut samples (**a**) and samples after tape stripping (**b**).

**Figure 13 polymers-14-03355-f013:**
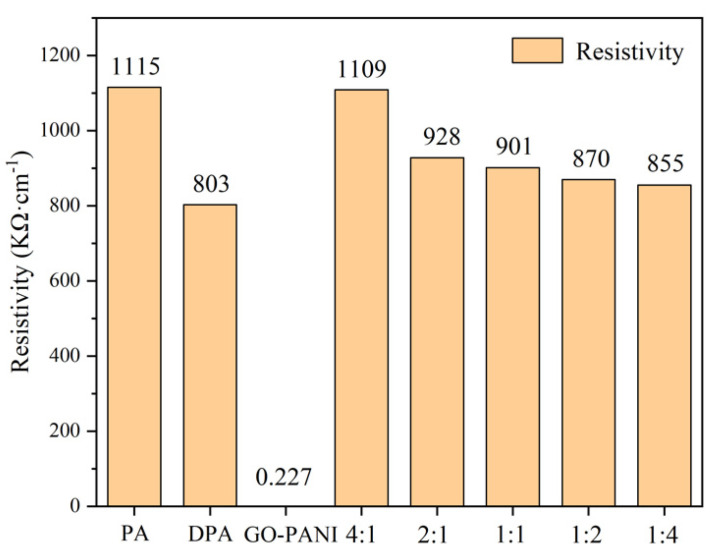
Conductivity of PA, PDA, GO-PANI and GO-PAN-PDA nanocomposites with different weight ratios.

**Figure 14 polymers-14-03355-f014:**
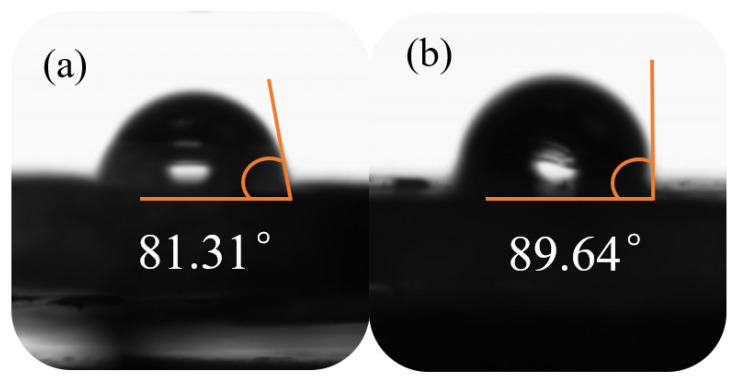
The contact angle results of (**a**) EP/PDA and (**b**) EP/GPP coating.

**Table 1 polymers-14-03355-t001:** The summarized Tafel plot data for the different coatings.

Sample	Eccor (V)	Iccor (A/cm^2^)	Corrosion Rate (mm/year)
EP/PDA	−0.873	1.78 × 10^−5^	2.08 × 10^−1^
EP/GP	−0.637	7.05 × 10^−7^	8.27 × 10^−3^
EP/GPP41	−0.737	9.90 × 10^−8^	1.16 × 10^−3^
EP/GPP21	−0.509	3.83 × 10^−8^	4.50 × 10^−4^
EP/GPP11	−0.550	3.92 × 10^−7^	4.60 × 10^−3^
EP/GPP12	−1.029	2.80 × 10^−6^	3.29 × 10^−2^
EP/GPP14	−0.810	1.12 × 10^−5^	1.31 × 10^−1^

## Data Availability

Data sharing is not applicable to this article.

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
