# Peer review of "Preparation and Characterization of Graphene Oxide/Polyaniline/Polydopamine Nanocomposites towards Long-Term Anticorrosive Performance of Epoxy Coatings"

_polymers, 2022, doi:10.3390/polym14163355_

Round 1

Reviewer 1 Report

This work reports an interesting update on the processing of polydopamine (PDA) coating, were GO-PANI-PDA nanocomposites successfully synthesized as an ideal filler to improve the anticorrosion properties of epoxy coatings. The reviewer has the following comments

In the introduction, the present background introduction about the epoxy resin, dopamine, and GO composites is not sufficient or accurate. Quite a few closely related topical articles on polymer composite especially thermal interface materials were unfortunately missing. Some latest comprehensive review articles citied and discuss such as: a) DOI: 10.1002/pat.3818 b) DOI: 10.1016/j.cej.2020.128301; d) DOI: 10.1002/aenm.202170135 and more other approaches through adding other particles into liquid metal etc. Please elaborate why weak bonding between PDA units will lead to weak bonding between PDA and substrates.

What is the relationship between the initial PDA coating thickness and the final thickness of salt spray experiments? Can the authors clarify this thickness change from 100 to 200 µm in when the salt spray process is first described?

Assuming a thickness change is an salt spray process, is there a minimum or maximum initial PDA thickness at which this slaty spray process may be applied? Also, it would be useful to discuss how the improvement in hardness and smoothness might change with the initial thickness of the PDA film.

Can the authors clarify how much the lack of scratch resistance for the regular PDA relates to the breaking free of the undesired PDA particle?

Good wear resistance does not guarantee good adhesion to substrates, which as stated by the authors is a weakness of PDA coating. Can the authors measure the adhesion between the coating and the substrate and make comparison to as-grown PDA?

Author Response

Response to Referee 1’s Comments

This work reports an interesting update on the processing of polydopamine (PDA) coating, were GO-PANI-PDA nanocomposites successfully synthesized as an ideal filler to improve the anticorrosion properties of epoxy coatings. The reviewer has the following comments

Comment 1. In the introduction, the present background introduction about the epoxy resin, dopamine, and GO composites is not sufficient or accurate. Quite a few closely related topical articles on polymer composite especially thermal interface materials were unfortunately missing. Some latest comprehensive review articles citied and discuss such as: a) DOI: 10.1002/pat.3818 b) DOI: 10.1016/j.cej.2020.128301; d) DOI: 10.1002/aenm.202170135 and more other approaches through adding other particles into liquid metal etc. Please elaborate why weak bonding between PDA units will lead to weak bonding between PDA and substrates.

Our Answer: Thank you very much for your comments. We cited the lastest comprehensive review articles as you suggested in references 1-3: a) DOI: 10.1002/pat.3818 b) DOI: 10.1016/j.cej.2020.128301; d) DOI: 10.1002/aenm.202170135. The adhesion between the PDA and the substrate is kind strong, because the catechol functional group of PDA has a strong metal coordination ability, which can promote the in-situ deposition of metal oxides coated on the surface. Such good interaction help improve the binding and stability between the coating layers and metal surface.

Comment 2. What is the relationship between the initial PDA coating thickness and the final thickness of salt spray experiments? Can the authors clarify this thickness change from 100 to 200 µm in when the salt spray process is first described ?

Our Answer: Thanks for your valuable comments. We confirmed that the thickness of the coating was not changed significantly during the salt spray experiment. Therefore, we added the decription in page 12 line 385, “It should be noted that we didn’t observe any significant thickness changes of coating during the salt spray experiments”. As we clarified in page 12 line 388, the earliest corrosion occurred in the left part of the plate in Fig. 12 (D7), and the corresponding corrosion time was 35 days with 100 μm thickness of EP/GPP21 coatings. Surprisingly, the surface with 200 μm thickness of EP/GPP21 coatings didn’t exhibit such corrosion, indicating that the 200 μm thickness of EP/GPP21 coatings has better anti-corrosive performance under the same salt spray testing conditions.

Comment 3. Assuming a thickness change is an salt spray process, is there a minimum or maximum initial PDA thickness at which this slaty spray process may be applied? Also, it would be useful to discuss how the improvement in hardness and smoothness might change with the initial thickness of the PDA film.

Our Answer: As we discussed in the last question, we confirmed that the thickness of the coating was not changed significantly during the salt spray experiment. Gerenally, the specific thickness of coatings hightly depends on the application needs, environment and construction conditions. In this study, we tried two different thickness (100 μm and 200 μm) and found that the earliest corrosion occurred in the left part of the plate shown in Fig. 12 (D7), and the corresponding corrosion time was 35 days with 100 μm thickness of EP/GPP21 coatings. Surprisingly, the surface with 200 μm thickness of EP/GPP21 coatings (Fig. 12, D8) didn’t exhibit such corrosion, indicating that the 200 μm thickness of EP/GPP21 coatings has better anti-corrosive performance under the same salt spray testing conditions.

This paper also measured toughness, contact angle, and wettability of the coating, which is closely related to the hardness and smoothness as some reference mentioned (such as, Graphene Oxide/Polyaniline Nanocomposites Used in Anticorrosive Coatings for Environmental Protection. Coatings, 2020,10(12): 1215; Graphene Oxide grafted with Dopamine as an efficient Corrosion Inhibitor for Oil Well Acidizing Environments. Surfaces and Interfaces, 2021,24: 101046; Bioinspired Ultrathin Graphene Nanosheets Sandwiched Between Epoxy Layers for High Performance of Anticorrosion Coatings. Chemical Engineering Journal, 2020,410: 128301.) The details were decribed in page 14 line 437, “The hydrophilicity and hydrophobicity of the coating surface were determined by contact angle in the presence of water. As shown in Figs. 15a and 15b, the contact angles of the EP/PDA coating and the EP/GPP coating were 81.31° and 89.64°, respectively, indicating that the hydrophobicity was greatly improved after the integration of PDA to GO-PANI. With the water-resistant barrier effect enhanced, the increased hydrophobicity of the EP/GPP coating could be beneficial for preventing the penetration of corrosive medium, leading to its good performance in the corrosion resistance test.”

Comment 4. Can the authors clarify how much the lack of scratch resistance for the regular PDA relates to the breaking free of the undesired PDA particle?

Our Answer: In the cross-cut test, it is necessary to use sufficient force to break the coating. Instead of scratch resistance, we pay more attention to the scratch on the surface of the substrate to see whether the nearby coating falls off, which will provide a strong estimation for the adhesion performance of the coating. In the 50 cm front and back impact performance tests, it was found that there was no cracking or peeling of the paint film on the front and back sides, indicating that the target coating had a good adhension. This is because the catechol group of the PDA coating has a strong metal coordination ability, which can promote the in-situ deposition of metal oxides under the coating on the surface of the material, and the good binding ability improves the interaction between the nano-hybrid material and the metal, giving rise to a reasonable stability. In page 12 line 406, we added the sentence “Next, to evaluate the adhesion performance of the coating, we applied sufficient force to completely break the coating and check the scratch on the surface of the substrate to see whether the nearby coating falls off.”

Comment 5. Good wear resistance does not guarantee good adhesion to substrates, which as stated by the authors is a weakness of PDA coating. Can the authors measure the adhesion between the coating and the substrate and make comparison to as-grown PDA?

Our Answer: The advantage of PDA incorporation is that it has good adhesion properties, as demonstrated in other literatures (Graphene Oxide grafted with Dopamine as an efficient Corrosion Inhibitor for Oil Well Acidizing Environments. Surfaces and Interfaces, 2021, 24: 101046). Although the wear resistance is an important parameter for the protective coating as you mentioned, the focus of this paper is to make a new type of coating filler consisting of graphene oxide/polyaniline/polydopamine (GO-PANI-PDA) nanocomposites with good anti-corrosion performance. Indeed, compared with GO-PANI, we demonstrated the incorporation of PDA into GO-PANI can significantly improve the adhesion, compatibility and anti-corrosion performance of the coating.

Reviewer 2 Report

In this article, preparation and characterization of graphene oxide/polyaniline /polydopamine nanocomposites and its long-Term anti-corrosive performance was mentioned by the authors  The manuscript looks good However, the following points should be address before publication.

Please check the spacing error throughout the manuscript and improve the quality of english language.

Need to mention why author choose polyaniline and polydopamine.

Authors need to mention a few recent references in introduction section,  in which same material has been used in diferent applications, https://doi.org/10.1016/j.matchemphys.2019.05.085,  Nanomaterials 12 (8), 1355, https://doi.org/10.1016/j.jtice.2018.07.041,  Polymers 14 (4), 845, Polymers 13 (1), 135,

Need to revise all blurred figures with high quality one.

EDX results should be incorporated for better understanding and interpretation.

Add one table regarding the comparison of previous studies and this study.

In results and discussion section, some critical analysis should be included.

The conclusion must be result oriented,  precise and attractive.

Author Response

Response to Referee 2’s Comments

Comment 1. Please check the spacing error throughout the manuscript and improve the quality of english language.

Our Answer: Thank you very much for your valuable comments. We have revised the manuscript for related issues.

Comment 2. Need to mention why author choose polyaniline and polydopamine.

Our Answer: In manuscript, page 2 line 55, we mentioned that “Polyaniline (PANI) is a well-known conductive polymer, which has light-weight, non-toxic, low-cost, good electrochemical and reversible redox characteristics[19-21]. Many reports have confirmed that PANI coating with a certain thickness has the effect on physical shielding and the special redox performance, which can promote the formation of a protective oxide film on the surface of metal and passivates the metal matrix[22, 23]. DeBerry et al.[24] found that PANI as an additive can alleviate the corrosion of metals. Schauer[25] and Zhang[26] added PANI to epoxy resin to form a good nanocomposite, where PANI can generate electrons to form a passivation film and form a chemical shielding layer.[27].”

In page 2 line 67, we highlighted “The amino and hydroxyl functional groups of Dopamine (DA) make it easy to be functionalized and cross-linked with other materials, resulting in the improvement of the dispersion ability and the compatibility between fillers and resins[28-33]. Beneficial from good compactness, anchorage and uniformity, the polydopamine (PDA) coating is a promising candidate filler for anti-corrosion coating[34, 35]. Zhu et al.[36] found that DA can enhance the compatibility and adhesion of graphene with epoxy resins and avoided galvanic corrosion. Haruna et al.[37] found that dopamine-functionalized graphene oxide (DA-GO) formed a protective layer to prevent carbon steel from corrosion. Due to the hydrogen bond between GO and PDA as well as the electrostatic interaction between -COO- in GO and -NH3+ in PDA, we propose that the grafting of dopamine on GO will help promote the compatibility of GO-PANI nanosheet in epoxy resin, thereby improving the corrosion inhibition efficiency of the coating”

Comment 3. Authors need to mention a few recent references in introduction section, in which same material has been used in diferent applications,  https://doi.org/10.1016/j.matchemphys.2019.05.085,  Nanomaterials 12 (8), 1355, https://doi.org/10.1016/j.jtice.2018.07.041,  Polymers 14 (4), 845, Polymers 13 (1), 135.

Our Answer: Your opinion is very important. After careful reading these articles, I found that they are very meaningful and helpful to this article. Some description was added and the references have been supplemented in the introduction part and cited as reference 38 and 39.

In page 2 line 78, we added “Nastaran et al. prepared a novel nanocomposite Ag-Pd-PDA/RGO to reduce the consumption of palladium, improve the catalytic activity of Pd, and have a synergistic effect. Bahram et al. confirmed that both barrier and inhibitive action of epoxy coating were significantly enhanced in the presence of GOQD-PANI. The fine GOQD-PANI well dispersed in the epoxy matrix, filled the pores and defects and blocked the diffusion pathways.” Besides, two references about GO-PANI “Polymers 14 (4), 845, Polymers 13 (1), 135.” were cited in reference 20 and 21.

 Comment 4. Need to revise all blurred figures with high quality one.

Our Answer: We carefully checked all figures. The new figures with better quality and resolution were replaced.

Comment 5. EDX results should be incorporated for better understanding and interpretation.

Our Answer: Thank you for your comments. Here we used a series of characterizations such as FTIR, XRD and SEM to demonstrate the successful synthesis of the GO-PANI-PDA nanocomposite. The relevant additional characterizations to aid understanding were provided in the supplementary file.

Comment 6. Add one table regarding the comparison of previous studies and this study.

Our Answer: Thanks so much for your valuable comments. In page 10 table 1, we made a summarized Tafel plot data for the different coatings of EP/GP, EP/PDA, EP/GPP14, EP/GPP12, EP/GPP11, EP/GPP21 and EP/GPP41. As we discussed in page 10 line 332, “Compared with the anodic and cathodic polarization curves of EP/GP coatings, the curves of EP/GPP coatings are shifted positively, which means that EP/GPP coating exhibits high corrosion potential during anodic and cathodic polarization. The cathodic and anodic branches are extrapolated to their intersections to obtain the corrosion current (Icorr), the corrosion potential (Ecorr) and the corrosion rate (CR)[51]. The corrosion potential of EP/GPP21 coating (-0.51 V) is shifted to the positive direction than that of EP/GP coating (-0.64 V). Its corrosion current density (3.83 × 10-8 A/cm2) is also nearly an order of mag-nitude lower than that of EP/GP (7.05 × 10-7 A/cm2), and the coating exhibits the smallest current density during cathodic polarization, corresponding to the corrosion rate of 4.50 × 10-4 mm/year is also an order of magnitude lower than that of EP/GP of 8.27 × 10-3 mm/year. These results indicated that the passivation zone was formed and EP/GPP21 coating has the best corrosion resistance. The similar results from Tafel analysis also suggested the same conclusion as that obtained by the EIS analysis.”

 Comment 7. In results and discussion section, some critical analysis should be included.

Our Answer: In page 6 line 242, we added the discussion about XRD patterns for GO, PANI and GO-PANI. And the corresponding XRD pattern was added in Fig. S1 in supporting information. In page 6 line 255, we added one more figure in Fig. S2 about the FT-IR spectra of GO for the better comparison. And the corresponding FTIR spectra was added in Fig. S2 in supporting information. In page 11 line 353, we added the decription “It is worth noting that the thickness of the electrode coating is very critical when making the electrode coating, and finding a more accurate coating thickness control method will be of great help to the experiment.”

Comment 8. The conclusion must be result oriented, precise and attractive.

Our Answer: We inserted more contects in conclusion parts. The changes have been highlighted as follows: “Doping GO-PANI in combination with the excellent properties of PDA makes GO-PANI nanosheets have better compatibility and dispersion in epoxy resins, strengthens the bonding performance between the coating and the substrate, and can also adjust the electrical conductivity of the material, fill the defects in EP/GP and improve the barrier properties and electrochemical passivation properties of GO-PANI. This work is expected to be widely used in the field of long-term corrosion protection, providing the necessary protection for various marine installations, ships and other metal components.”

Round 2

Reviewer 1 Report

The authors have addressed my concerns and the paper is now acceptable for publishing.